# The Intention of Community Participation in the Qilian Mountain National Park Policy Pilot

**Liqi Jia [1], Junqing Wei [1] and Zibin Wang [2],***

1. School of Design Art, Lanzhou University of Technology, Lanzhou 730050, China; liqijia@lut.edu.cn (L.J.);
   JunqingWei@lut.edu.cn (J.W.)
2. School of Educational Science and Technology, Northwest Minzu University, Lanzhou 730030, China
* Correspondence: wang-zibin@xbmu.edu.cn

**Abstract:** As a management strategy, community participation is to implement the coordinated development of communities and protected areas. In recent years, the development of China's national parks has faced many challenges related to human and environmental constraints. Community participation plays an essential role in solving such issues. As one of the critical indicators to test community participation, community residents' willingness to participate significantly impacts community participation in constructing national parks. As such, this study was conducted using the extended model of the theory of planned behavior (TPB) and the structural equation model. Taking the Tianzhu county and Sunan Yugu county as examples, and based on 230 valid questionnaires, we investigated the impacts of the Qilian Mountain National Park System Pilot Area on community residents' willingness to participate and provided relevant suggestions for amendments. The results indicated that, for the Qilian Mountain National Park System Pilot Area, behavioral attitude, subjective norms, and perceptual behavior control positively impacted the participation intention of community residents. At the same time, the variables mentioned above positively impacted the implementation of the participation intention of community residents. Specifically, the order of impacts is as follows: perceptual behavior control (path coefficient = 0.89) > participation behavior attitude (path coefficient = 0.68) > related impact system (path coefficient = 0.41) > subjective norms (path coefficient = 0.38). According to the results, we put forward three suggestions: (1) providing relevant instructions and guidance on various methods to ensure that the pilot policies on the construction of national parks can form a positive relationship with the participation intentions of the community residents; (2) making full use of the function of perceptual behavior control, so the subjective initiative of community residents can be maximized, thereby enhancing the willingness of community residents to participate in constructing national parks; and (3) strengthening the impacts of subjective norms, enhancing the soft culture of national park communities' participation, reshaping the community cultural landscapes with the goal of constructing national parks, and establishing community residents' sense of honor as the builders of national parks.

**Keywords:** Qilian Mountain National Park; community participation; TPB extended model; balloon dessert; structural equation model





## 1. Introduction

The mechanism of community participation in national parks originated in the United States in the 1960s and 1970s [1] as an autonomous management policy that includes community residents. After a long period of evolution [2,3], it has become one of the most crucial components of the management measures of protected areas. The core idea of this concept is to encourage the residents of national park communities to participate in the construction of national parks to varying degrees as participants and beneficiaries of the national park system [4]. The majority of international research on community participation has focused on the macro level [5]. Most of this research was carried out through

semi-structured interviews and analyzed individual cases or multiple cases [6]. The major factors that affect the willingness of a community's participation include management system issues, implementation progress of relevant policies, ecological compensation, and related conflicting interests [7,8]. At present, China's national parks have adopted the same policy guidance [9]; meanwhile, it reviews and compares the community participation policies implemented by the protected areas and explores the different issues encountered by the community participation policies in developing China's nature reserves [10]. In the current research related to the willingness to participate in national parks, the primary evaluation is based on measuring the perceived value of community residents [11]. As major events affecting the community, National Park System Pilot Areas have strengthened local residents' sense of place and identity [12,13], becoming one of the core elements that affect the willingness of the participation of community residents. At the same time, some scholars have claimed that the mechanism of national park community participation relies primarily on cognition, attitude, and participation [14]. Mensah believes that the impact of the perceived financial benefits of tourism has a significant influence on community engagement [15]. However, other scholars hold different attitudes toward relevant conclusions based on different research areas. These researchers claim that community engagement in tourist development is strongly dependent on gatekeepers' attitudes and communities' economic backgrounds. Ensuring community participation is more difficult in settings where economically vulnerable communities and manipulative gatekeepers are present. As a result, sustainable land and resource use practices are hindered, resulting in irreparable damage to environmentally sensitive areas [16].

Community participation is an essential part of the development of national parks. The willingness to participate is a prerequisite for the effective implementation of the mechanism of community participation; it is mostly affected by the behavioral attitudes and perceived value of community residents. Therefore, this study was conducted using the theory of planned behavior (TPB), which Ajzen proposed in 1977 [17]. This model is used to explain the behavioral attitude and behavioral intention of the research object, and a large number of scholars have used this model to study problems related to national parks. Miller used the TPB model theory to investigate and analyze the phenomenon of human–animal conflict among tourists in Yellowstone National Park in the United States and put forward practical suggestions for tourists and managers to prevent such incidents [18]. Goh et al. (2017) studied tourists' intentions to go off-trail in the Blue Mountains National Park (BMNP) in Australia and revealed that pro-environmental attitudes effectively predict general environmental worldviews [19]. Using the theory of planned behavior, Reigner et al. (2009) analyzed the relationships among visitors' attitudes, subjective norms, and perceived control over pool exploration, intentions to explore, and actual actions at pools [20]. This theoretical model is mainly used to explain the behaviors, attitudes, and intentions of the research objects. Nevertheless, the adaptability of some research subjects can hinder analysis when the model is applied to research in various fields. Behavioral intentions are often restricted by objective conditions. Ajzen also recognized the existence of such an issue and therefore pointed out that corresponding corrections or extensions are needed when utilizing the TPB theoretical model [1] in order to better adapt to research subjects when applied to different disciplines [21]. In the current research, Wenbin Zhang introduced ecological compensation mechanisms as an extension of the TPB model and analyzed the willingness for ecological protection and behavioral intentions of the residents in environmentally protected areas [22]. Han tested the established TPB model and explained consumers' behavioral intentions with regard to choosing eco-friendly hotels by using the structural equation model [23]. Yuangang Zhang introduced local theory into the TPB extended model as a research variable to analyze the impact of local emotions on tourists' traveling behaviors [24]. Using the theory of planned behavior (TPB), Wang et al. (2019) investigated the effect of the EB of a tourist spot on the ERB using a structural equation model (SEM) multi-group study (MGA) [25]. For hikers visiting a national park in Taiwan, Wang et al. (2020) investigated a behavioral model employing the latent variables

of personality, environmental concern, attitudes toward activities, and environmentally responsible behavior [26]. This study will also introduce extended variables when using the theory of planned behavior (TPB): the impact of the pilot implementation of Qilian Mountain National Park on the willingness to participate of community residents (the following is the relevant impact system).

Based on the theory of planned behavior (TPB), this paper constructs a theoretical model in five aspects, including pilot implementation of national park policies, residents' behavioral attitudes, subjective norms, perceived behavioral control, and the willingness of residents' community participation. Taking the Gansu area of Qilian Mountain National Park as an example, this study focused on using a structural equation model to investigate the mechanism of the influence of policy pilots on community residents' willingness to participate in the construction of national parks. The innovations of this article include the following: (1) taking the perspective of collective choice as a prerequisite, exploring the changes in the degree of recognition of community participation through residents' subjective willingness in the circumstances of the implementation of major policies; (2) using the National Park System Pilot Area as the only influential factor among independent variables to explore whether community participation is affected, deepening the understanding of the path of influence based on the theory of planned behavior (TPB); and (3) when investigating the impact of intermediate variables (behavioral attitudes, subjective norms, perceptual behavior control) on the willingness for community participation through national park policy pilots, using differentiated analysis of the path coefficients of three intermediate variables, to a certain extent, clarifying the core influential conditions and deficiencies regarding community residents' willingness to participate and providing targeted solutions for future community participation in the construction of national parks.

## 2. Materials and Methods

This work aimed to investigate the various impacts on the willingness of indigenous communities to participate with regard to the Qilian Mountain National Park Pilot Area. We first made assumptions regarding the corresponding research based on the research framework. The data results were then screened using questionnaire surveys, the Delphi method, and semi-structured interviews. Finally, we analyzed the preset results and drew reasonable conclusions through the use of structural equation modeling (SEM).

### 2.1. The Construction of the Theoretical Model

The theory of planned behavior model was used to explain the subjective willingness of the research participants. It contains three specific stages, including the perceptional stage, attitude cognition, and behavioral intention. Existing research has analyzed residents' perceptions of the relevant measures implemented in the management regions after using the National Park System Pilot Area [27]. The stage of attitude cognition is mainly due to the subjective judgment made by the research participants after being affected by external influences [28]. Behavioral intention is primarily the subject of the dependent variables. Existing research has focused mainly on investigating the willingness to ensure environmental protection and tourism intention [24]. The following is the research framework (based on the theory of planned behavior) used in this article (see Figure 1) [2].

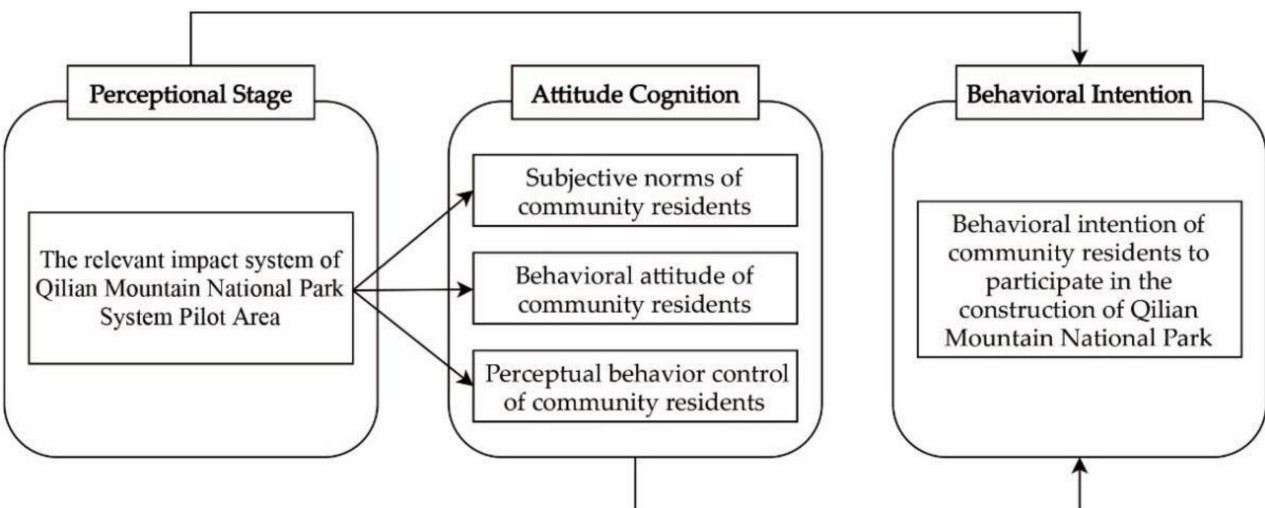

**Figure 1.** Extended model.

*2.2. Theoretical Hypothesis*

The perception element is the Qilian Mountain National Park System Pilot. As the core component of the extended proposition, it will increase the popularity of the Qilian Mountain community to a certain degree; meanwhile, the community will also become a crucial supporting point regarding implementing the functionality of the national park. Against the background of the implementation of the pilot policy, community residents' intention to participate will be affected by their subjective awareness and the external objective conditions [23]. Therefore, the hypotheses related to the relevant impact of the Qilian Mountain National Park System Pilot Area are as follows:

- H1: the relevant impact system of the Qilian Mountain National Park System Pilot Area has a positive and significant influence on the behavioral intentions to participate of its community residents.

  - $H_{1-1}$: the relevant impact system of the Qilian Mountain National Park System Pilot Area has a positive and significant influence on the behavioral attitudes toward participation of its community residents.

  - $H_{1-2}$: the relevant impact system of the Qilian Mountain National Park System Pilot Area has a positive and significant influence on the subjective norms of its community residents.

  - $H_{1-3}$: the relevant impact system of the Qilian Mountain National Park System Pilot Area has a positive and significant influence on the perceived behaviors of its community residents.

Attitudes and behaviors are related [23]; therefore, attitude is defined as a certain cognitive tendency and is restricted by an individual's perception and preference [24]. This article is based on the relevance of the Qilian National Park System Pilot Area and the related impact system on community residents. When community residents recognize that they benefit from the National Park System Pilot Area, their behavioral attitudes will be more positive, which has a positive impact regarding the willingness of the community to participate, and vice versa. Thus, the relevant hypothesis on behavioral attitude includes

- H2: the behavioral attitude of the participation of community residents in the Qilian Mountain National Park System Pilot Area has a positive and significant influence on the community's intention of participation.

Subjective norms can be defined as the external impetus given by others when an individual completes or performs a particular task or behavior. This impetus can be a positive expectation or negative pressure. In addition, subjective norms reflect an individual's

desire to receive relevant support and approval from the public when performing a certain behavior [27,28]. Therefore, the relevant hypothesis regarding the subjective norms is:

- H3: the subjective norms of community residents in the Qilian Mountain National Park System Pilot Area have a positive and significant influence on the behavioral intentions of the community's participation.

Based on the idea of perceptual behavior control, individuals are restricted by both external and internal factors during the process of conducting some kind of behavior. In most situations, individuals' judgment about something is often restricted by their intellectual level, recognition, and external factors, rather than being based on a sense of objectivity and rationality. Therefore, the hypothesis related to perceptual behavior control is:

- H4: the perceptual behavior control of community residents in the Qilian Mountain National Park System Pilot Area has a positive and significant influence on the behavioral intentions of the community's participation.

### 2.3. Research Methods

Based on the existing hypothesis and present theoretical model, we collected relevant data using a questionnaire survey and analyzed the obtained data comprehensively and linearly using the structural equation model (SEM). Reasonable amendments were then made to the relevant hypothesis, finally obtaining a convincing model for the National Park System Pilot Area.

The study contains two sections. The first section details the survey design and is divided into three stages: proposition of the hypothesis, expert consultation, and prior observation. As the core part among the three stages in the survey design, the proposition of the hypothesis divided the present hypothesis into three categories. The first category is related to participants' cognition. We utilized existing research to establish a hypothesis regarding the impact of major events on the perception of community residents. The second category concerns participants' attitudes. We referred to the propositional research conducted by Yuangang Zhang [24] and Wenbin Zhang [22] to design the research hypothesis on three dimensions, including the subjective norms, the behavioral attitudes, and the perceptual behavior control of community residents. The third category involves investigating participants' behavioral intentions. We proposed the hypothesis based on relevant studies conducted by Qunming Zheng [29] and collected data using the Likert scale (1 = strongly agree, 2 = agree, 3 = slightly agree, 4 = disagree, 5 = strongly disagree).

The second section includes data analysis. Specifically, the authors conducted a component matrix rotation and tested the reliability and validity of the obtained data. After delimiting the component intervals, the structural equation model (SEM) was used to perform further tests. Because the dependent variable in this study is unobservable, and the independent variable affects multiple intermediate variables simultaneously, compared to other testing models, the structural equation model (SEM) testing method is linear and can predict the relationships of multiple interrelated variables simultaneously. In addition, it allows researchers to cope with unobservable variables in parts of models, and it helps to explain the measurement errors in the overall estimation process. The authors later referred to the study conducted by Hair et al. (1998) [30] and found that it is more reasonable to require the ratio of items to the sample number of the model to be 1/10 to 1/15 [31].

## 3. Data Collection and Analysis

Based on the previously detailed hypotheses and model construction, this study collected data from two gateway communities of Qilian National Park. Descriptive analysis of related variables and normal distribution tests was used to analyze the collected data.

### 3.1. Description of the Research Area

The research areas in this study includes Sunan county and Tianzhu county in Qilian Mountain National Park (see Figure 2). Both counties are important gateway communities

of the Qilian Mountain National Park. Specifically, Sunan county is located in the middle of the Hexi Corridor and the north of the Qilian Mountain, with a total area of 23,800 km$^2$ (2014). With a relatively large proportion of indigenous people in the community, Sunan county has a total population of 37,579. In addition, the Sunan Yugu nationality is a unique ethnic minority group in Gansu Province. Tianzhu county is located at the eastern end of the Hexi Corridor, specifically on the northeastern edge of the Qinghai-Tibet Plateau. It is known as the "gateway" to the Hexi Corridor and borders Sunan in the northwest. Before implementing the National Park System Pilot Area, these two counties' development and industrial structures were mainly resource-oriented, including tourism, animal husbandry, and plantations. In addition, the production methods of these two counties were relatively backward, and the multi-ethnic settlements were the main form of distribution. The implementation of the National Park System Pilot Area has had a major impact on these two areas; thus, they are appropriate for the research.

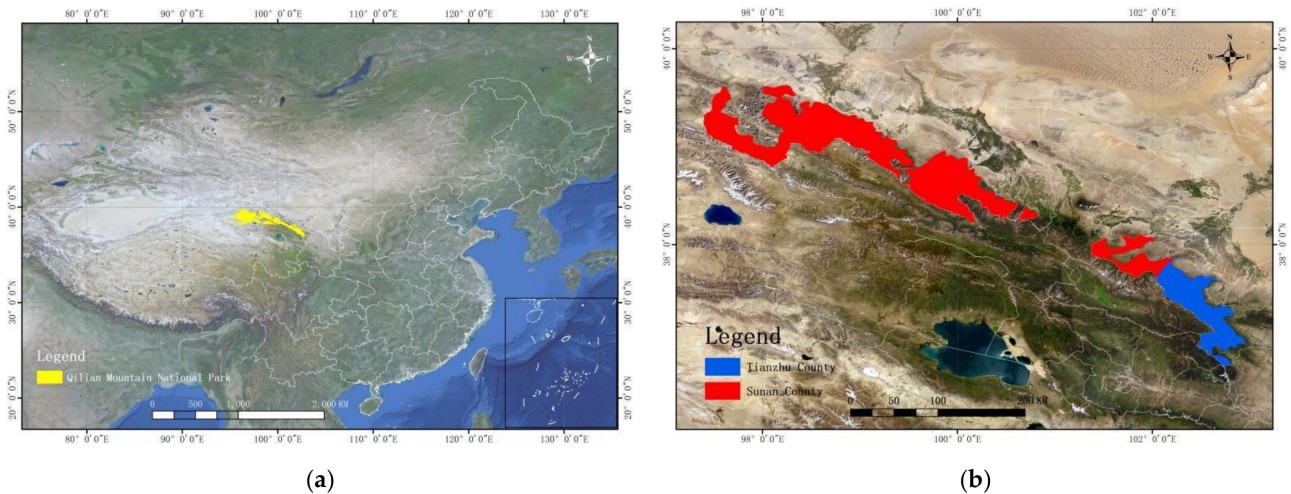

(**a**)　　　　　　　　　　　　　　　　　　　　　　　　　　　　　(**b**)

**Figure 2.** The location of Qilian Mountain National Park, Tianzhu county and Sunan county: (**a**) the location of Qilian Mountain National Park in China; (**b**) the location of Tianzhu county and Sunan County in Qilian Mountain National Park.

### 3.2. Demographic Sample Analysis

The distribution of the questionnaires comprised two methods: field distribution and online collection. The research team went to Sunan county and Tianzhu county to distribute questionnaires on site from 12 to 15 April 2020 and 1 to 3 June 2020. The online questionnaire was conducted in the form of a mobile phone app.

The survey took the family as the basic unit and selected one person from each household as the survey sample. After informing the respondents of their relevant survey obligations and obtaining permission, a one-to-one household survey was conducted. The sample selection mainly relied on the theoretical sampling method [32]. The specific number of people was determined according to the community population and industry data. The community committee recommended the industry representative family. To ensure the uniform spatial distribution of the samples and to include all representative industries, we classified the survey based on investigating the distance between household samples of different representative industries and the core areas of the national park, as well as the distance between the residence of different samples and the core area of the national park.

A total of 160 questionnaires were collected during the first distribution. After excluding invalid questionnaires, the actual valid samples totaled 147, with a 91.9% return rate. In addition, a second round of questionnaire distribution was carried out through an online collection method. Researchers distributed 90 questionnaires, and 83 valid samples were collected. The return rate was 92.2%.

Most respondents were between 26 and 45 years old, accounting for 47.8%; the educational background of respondents was mostly secondary school and below, accounting for 80.5%; and the monthly income of the majority of respondents was below 6000 RMB, accounting for 79.5%. In addition, most residents of the community were born locally, accounting for 64.3% of respondents. Moreover, the research areas of this study include minority autonomous regions, with agriculture and animal husbandry as the pillar industries, accounting for 40% of all sources of income (see Table 1).

**Table 1.** Demographic sample.

| Survey Item | Type | Frequency (Sample = 230) | Percentage (%) |
|---|---|---|---|
| Gender | Male | 120 | 52.2 |
| | Female | 110 | 47.8 |
| Age | Under 16 | 4 | 1.7 |
| | 16–25 | 57 | 24.8 |
| | 26–45 | 110 | 47.8 |
| | 45–65 | 47 | 20.4 |
| | Over 65 | 12 | 5.3 |
| Education | Primary school | 16 | 7 |
| | Junior school | 66 | 28.7 |
| | High school | 92 | 40 |
| | College | 30 | 13 |
| | Undergraduate | 13 | 5.7 |
| | Postgraduate | 2 | 0.8 |
| | Other | 11 | 4.8 |
| Average Salary per Month | Less than 1000 (RMB) | 7 | 3 |
| | 1001–3000 (RMB) | 60 | 26.1 |
| | 3001–6000 (RMB) | 116 | 50.4 |
| | 6001–9000 (RMB) | 41 | 17.8 |
| | Over 9001 (RMB) | 6 | 2.7 |
| Attribute of Residents | Native Settlers | 148 | 64.3 |
| | Migrants (Non-Native Settlers) | 82 | 35.7 |
| Source of Income | Tourism | 70 | 30.4 |
| | Animal Husbandry and Plantation | 95 | 41.3 |
| | Other | 65 | 28.3 |
| Distance from the Core Region | Less than 5 km | 37 | 16.1 |
| | 5–10 km | 55 | 23.9 |
| | 10–15 km | 86 | 37.4 |
| | 15–20 km | 52 | 22.6 |

*3.3. Descriptive Analysis*

Integrating the effective questionnaires (see Tables 2 and 3), we used SPSS 23.0 to perform a descriptive analysis of the distribution patterns of the mean, standard deviation, variance, skewness, and kurtosis of the survey sample items. The questionnaire in this study contained 19 items. We divided the 19 items into five potential variables according to the dimensions, including related impact system, behavior attitude, subjective norms, perceptual behavior control, and behavior intention. The items were investigated in the form of a Likert-5 scale, where the numbers 1, 2, 3, 4, and 5, respectively, represented strongly agree, agree, slightly agree, disagree, and strongly disagree. Regarding the results, the standard deviation of each item was greater than 0.60; the Likert-5 scale thus corresponded with the research expectations. The skewness value was between −0.974 and −0.298, the kurtosis value was between −0.531 and −0.091, the absolute value of the skewness was

less than 3, the absolute value of the kurtosis was less than 10, and the sample data were normally distributed.

**Table 2.** Research aspects and measurement items.

| Research | Measurement | Hypothesis | References |
|---|---|---|---|
| Related Impact System Scale of Qilian Mountain National Park System Pilot Area | Stage of Cognition | RIS1: I have a basic understanding of the related information of the Qilian Mountain National Park System Pilot Area and the communities' functions regarding the construction of the pilot area.<br>RIS2: I understand the policy mechanism implemented in the Qilian Mountain National Park System Pilot Area.<br>RIS3: The government actively promoted relevant knowledge to community residents in constructing the Qilian Mountain National Park System Pilot Area.<br>RIS4: The government provided policy guidance and technical support in constructing the Qilian Mountain National Park System Pilot Area. | Zhang et al. (2017) [22]<br>Zhou et al. (2017) [27] |
| Behavioral Attitude Scale of Qilian Mountain National Park Residents' Participation | Attitude Cognition | BA1: The construction of the Qilian Mountain National Park is inseparable from the participation of the community, which is the core element of the development of the national park.<br>BA2: The Qilian Mountain National Park implements a community co-management mechanism, which is also the future trend regarding the development of national park communities.<br>BA3: The Qilian Mountain National Park System Pilot Area can generate revenue based on eco-tourism and I can profit from it.<br>BA4: The Qilian Mountain National Park System Pilot Area can increase awareness of my community and increase individuals' sense of pride. | Han et al. (2010) [23]<br>Zhang et al. (2017) [24] |
| Subjective Norms Scale of Qilian Mountain National Park Residents | Attitude Cognition | SN1: The Qilian Mountain National Park Administration believes that community residents' awareness of participation in the construction of national parks should be raised at this stage.<br>SN2: Schools and relevant education departments believe that community residents' awareness of participation in the construction of national parks should be raised at this stage.<br>SN3: My friends and family members believe that community residents' awareness of participation in the construction of national parks should be raised at this stage | Zhou et al. (2014) [28] |
| Perceptual Behavior Control Scale of Qilian Mountain National Park Residents | | PBC1: I have a basic understanding of the process of the community participation and related policies for the Qilian Mountain National Park System Pilot Area.<br>PBC2: I can take on relevant responsibilities as a community resident after implementation of the Qilian Mountain National Park System Pilot Area.<br>PBC3: I have an optimistic attitude towards the intentions of community residents' participation after implementation of the Qilian Mountain National Park System Pilot Area.<br>PBC4: I have a supportive attitude towards community residents' active participation in constructing the Qilian Mountain National Park System Pilot Area. | Wang et al. (2020) [31] |
| Behavioral Intention Scale of Qilian Mountain National Park Residents' Participation | Behavioral Intention | BI1: As a community resident, I am willing to actively participate in constructing the Qilian Mountain National Park.<br>BI2: I will actively cooperate with the National Park Administration to fulfill various requirements for community construction.<br>BI3: I will encourage people around me to participate in the project actively and ask them to learn relevant information.<br>BI4: I will actively participate in the volunteer activities needed in the construction of the national park. | Zheng et al. (2014) [29] |

**Table 3.** Normal distribution data test.

| Abbreviations of Measurement Hypothesis | Mean | Standard Deviation | Variance | Skewness | | Kurtosis | |
|---|---|---|---|---|---|---|---|
| | Statistics | Statistics | Statistics | Statistics | Standard Error | Statistics | Standard Error |
| $RIS_1$ | 3.6478 | 1.14166 | 1.303 | −0.644 | 0.16 | −0.13 | 0.32 |
| $RIS_2$ | 3.5565 | 1.12682 | 1.303 | −0.595 | 0.16 | −0.2 | 0.32 |
| $RIS_3$ | 3.4652 | 1.19902 | 1.438 | −0.638 | 0.16 | −0.261 | 0.32 |
| $RIS_4$ | 3.8565 | 1.18996 | 1.416 | −0.974 | 0.16 | 0.133 | 0.32 |
| $BA_1$ | 3.513 | 1.21033 | 1.465 | −0.56 | 0.16 | −0.531 | 0.32 |
| $BA_2$ | 3.2217 | 1.12473 | 1.265 | −0.298 | 0.16 | −0.451 | 0.32 |
| $BA_3$ | 3.4348 | 1.16821 | 1.365 | −0.571 | 0.16 | −0.227 | 0.32 |
| $BA_4$ | 3.5 | 1.16255 | 1.352 | −0.538 | 0.16 | −0.28 | 0.32 |
| $SN_1$ | 3.7087 | 1.20675 | 1.456 | −0.717 | 0.16 | −0.316 | 0.32 |
| $SN_2$ | 3.6348 | 1.15468 | 1.333 | −0.625 | 0.16 | −0.294 | 0.32 |
| $SN_3$ | 3.613 | 1.20146 | 1.443 | −0.696 | 0.16 | −0.177 | 0.32 |
| $PBC_1$ | 3.5783 | 1.1599 | 1.345 | −0.7 | 0.16 | −0.091 | 0.32 |
| $PBC_2$ | 3.5304 | 1.15082 | 1.324 | −0.621 | 0.16 | −0.203 | 0.32 |
| $PBC_3$ | 3.8957 | 1.0645 | 1.133 | −0.929 | 0.16 | 0.377 | 0.32 |
| $PBC_4$ | 3.6565 | 1.14812 | 1.318 | −0.642 | 0.16 | −0.344 | 0.32 |
| $BI_1$ | 2.4652 | 1.29699 | 1.682 | 0.647 | 0.16 | −0.663 | 0.32 |
| $BI_2$ | 2.5565 | 1.3262 | 1.682 | 0.37 | 0.16 | −1.076 | 0.32 |
| $BI_3$ | 2.2696 | 1.25953 | 1.586 | 0.868 | 0.16 | −0.304 | 0.32 |
| $BI_4$ | 3.2826 | 1.4059 | 1.977 | −0.38 | 0.16 | −1.089 | 0.32 |

## 4. Results

After analyzing the variables using descriptive analysis, we utilized Cronbach's alpha to test the reliability and validity of the measurement indicators of the questionnaire. In addition, the maximum likelihood estimation was used to analyze the structural model after the reliability and validity met the basic research requirements.

### 4.1. Reliability Analysis

We performed the reliability analysis using SPSS 23.0 to test 19 observable variables in 230 returned questionnaires. In most situations, the ideal Cronbach's alpha coefficient of a scale should be above 0.6. This study's overall Cronbach's alpha coefficient was above 0.8, which means that the obtained survey results had good internal consistency (see Table 4).

**Table 4.** Reliability analysis.

| Item | Cronbach's Alpha | Standardized Cronbach's Alpha | Number of Items |
|---|---|---|---|
| Overall Scale | 0.846 | 0.858 | 19 |
| Related Impact System Scale | 0.871 | 0.871 | 4 |
| Behavioral Attitude Scale | 0.886 | 0.886 | 4 |
| Perceptual Behavior Control Scale | 0.855 | 0.856 | 3 |
| Subjective Norms Scale | 0.817 | 0.819 | 4 |
| Behavioral Intention Scale | 0.879 | 0.883 | 4 |

Reliability analysis is a standard method to test the validity of a survey. It measures how the sample data reflects the final research contents and goals. Therefore, the higher the value of reliability, the more the survey data reflects the authentic results of the research. In general, there are two types of reliability analysis: content validity and structure validity. Because the content of the items involved in this study was reviewed and analyzed by experts in the relevant field, we did not focus on the content validity in this article.

We conducted correlation tests on the Kaiser-Meyer-Olkin (KMO) value and Bartlett sphericity. The results (see Table 5) indicated that the KMO value was greater than 0.8; and the sphericity test value was 2519. The degree of freedom (Sig) was less than 0.001 (generally, it is reasonable to conduct factor analysis when the value of KMO is greater than 0.8 and the Sig value is less than 0.001), so it was suitable for subsequent factor analysis. In terms of structure validity, we extracted and analyzed the factors through principal component analysis. According to the existing model, we extracted and rotated five common factors using the Kaiser maximum variance method. The results in Table 6 show A1 is highly correlated with RIS1–4, A2 is highly correlated with BA1–4, A3 is highly correlated with BI1–4, A4 is correlated with SN1–3, and A5 is highly correlated with PBC1–5. Combined with the results displayed in Table 7, it was found that the cumulative contribution rate of the total variance of the load factor was 71.303%, which is greater than 60% (it is generally considered that the survey has good structural validity when the cumulative contribution rate of the total variance of the load factor is greater than 0.6), so the questionnaire used in this study had good validity.

**Table 5.** KMO value and Bartlett sphericity test.

| Scale Type | KMO Sampling Suitability Quantity | Bartlett Sphericity Test | | |
| --- | --- | --- | --- | --- |
| | | Approximate Chi Square | Degree of Freedom | Significance |
| Overall scale | 0.93 | 2519.525 | 171 | 0.0000 |
| Related Impact System Scale | 0.831 | 442.137 | 6 | 0.0000 |
| Behavioral Attitude Scale | 0.824 | 509.384 | 6 | 0.0000 |
| Conceptual Behavior Scale | 0.822 | 597.195 | 6 | 0.0000 |
| Subjective Norms Scale | 0.85 | 638.718 | 6 | 0.0000 |
| Behavioral Intention Scale | 0.825 | 916.342 | 6 | 0.0000 |

**Table 6.** Composition matrix after rotation.

| Items | Composition | | | | |
| --- | --- | --- | --- | --- | --- |
| | A1 | A2 | A3 | A4 | A5 |
| RIS2 | 0.769 | | | | |
| RIS4 | 0.755 | | | | |
| RIS3 | 0.722 | | | | |
| RIS1 | 0.716 | | | | |
| BA1 | | 0.798 | | | |
| BA2 | | 0.773 | | | |
| BA3 | | 0.734 | | | |
| BA4 | | 0.642 | | | |
| BI1 | | | −0.874 | | |
| BI2 | | | −0.855 | | |
| BI3 | | | −0.732 | | |
| BI4 | | | 0.569 | | |
| SN3 | | | | 0.764 | |
| SN2 | | | | 0.749 | |
| SN1 | | | | 0.745 | |
| PBC3 | | | | | 0.679 |
| PBC4 | | | | | 0.611 |
| PBC2 | | | | | 0.6 |
| PBC1 | | | | | 0.579 |

**Table 7.** Variance contribution rate of load factor.

| Composition | Initial Eigenvalue | | | Sum of Squares of Rotating Load | | |
|---|---|---|---|---|---|---|
| | Total | Percentage of Variance | Cumulation (%) | Total | Percentage of Variance | Cumulation (%) |
| A1 | 8.603 | 45.276 | 45.276 | 3.092 | 16.276 | 16.276 |
| A2 | 1.947 | 10.249 | 55.525 | 2.965 | 15.605 | 31.88 |
| A3 | 1.207 | 6.353 | 61.878 | 2.856 | 15.031 | 46.911 |
| A4 | 0.921 | 4.849 | 66.726 | 2.444 | 12.863 | 59.774 |
| A5 | 0.869 | 4.576 | 71.303 | 2.19 | 11.528 | 71.303 |

### 4.2. Structural Equation Model Analysis

Based on the pre-mentioned research method, this researchers in this study used the structural equation model (SEM) to perform a linear analysis on the established extended model of the Theory of Planned Behavior (see Figure 1) to verify whether the positive relationships between variables is reasonable.

### 4.2.1. Parameter Fitting Analysis

It is necessary to conduct a parameter fitting analysis when utilizing the structural equation model to do research. The leading indicators contain the chi-square value (CMIN) and the degree of freedom value (DF); a ratio of these values between 1 and 3 indicates the model has a good degree of fitting. In addition, if the value of the root mean square error of approximate (RMSEA) ranges from 0.05 to 0.08, this indicates that the result is reasonable; a value less than 0.05 means that the degree of fitting is better. For IFI, NFI, AGFI, CFI, and RFI (incremental fit index, normed fit index, adjusted goodness of fit index, comparative fit index, and relative fix index, respectively), the range is usually between 0 and 1. Specifically, the closer the index is to 1, the better the fit. If the index is greater than 0.9, it indicates a good degree of fit (see Table 8).

**Table 8.** Model fitting index.

| CMIN/DF | RSMEA | IFI | NFI | GFI | AGFI | CFI | RFI |
|---|---|---|---|---|---|---|---|
| 1.697 | 0.055 | 0.959 | 0.959 | 0.899 | 0.968 | 0.958 | 0.938 |
| pass | good | pass | pass | acceptable | pass | pass | pass |

### 4.2.2. Hypothetical Test

Commonly, the hypothesized model is tested after the fitting analysis. Generally, the relationship between the absolute value of the standardized path coefficient and the variable is positively correlated. The positive and negative values represent the relevant influence directions. At the same time, the critical ratio (CR) must be satisfied when the absolute value is larger than 1.96 and the $p$-value is less than 0.05. When these conditions are met, the hypothesis is supported.

In this study, we analyzed the related variables of the hypothesized structural equation model based on the maximum likelihood method. The standardized path coefficients of the analyzed variables had a high significance level, and they met the relevant requirements of the CR and $p$-value. After verification, it was found that the hypotheses were all true (see Table 9). The specific analysis is as follows (see Figure 3).

**Table 9.** Model hypothesis test.

| Path | Estimate | SE | CR | p | Result |
|------|----------|-----|-----|-----|--------|
| BA<—RIS | 0.706 | 0.077 | 9.199 | *** | Support |
| SN<—RIS | 0.862 | 0.081 | 10.694 | *** | Support |
| PBC<—RIS | 0.674 | 0.078 | 8.629 | *** | Support |
| BI<—RIS | 0.413 | 0.292 | 4.714 | *** | Support |
| BI<—BA | 0.676 | 0.136 | 4.738 | *** | Support |
| BI<—SN | 0.38 | 0.149 | 3.257 | 0.037 | Support |
| BI<—PBC | 0.892 | 0.332 | 2.388 | 0.017 | Support |

Note: *** represents strong statistical significance and the value is 0.001.

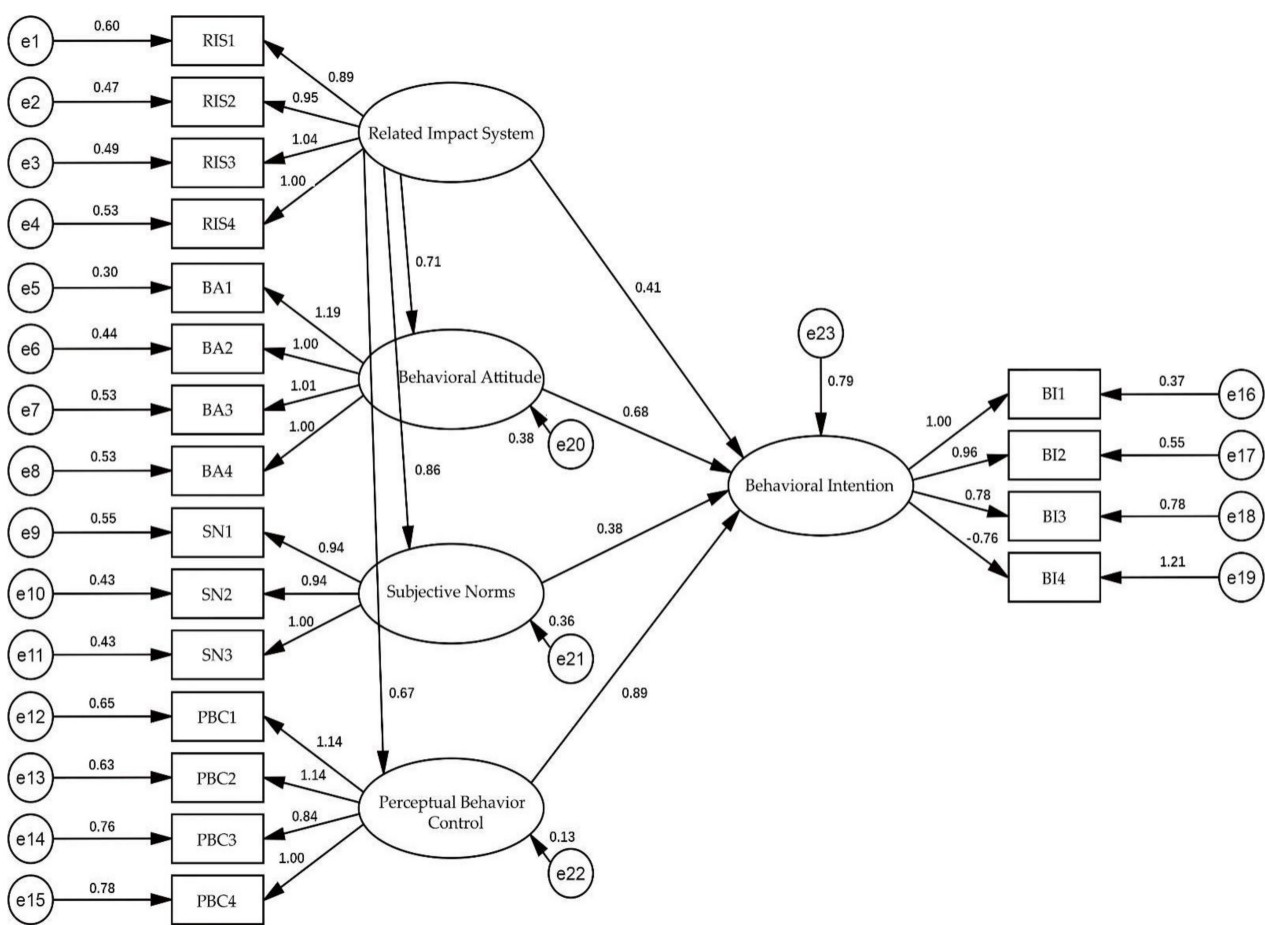

**Figure 3.** Structural equation model.

The critical ratio (CR) of the relevant influence system (RIS) on community residents' behavioral intention (BI), behavioral attitude (BA), subjective norms (SN), and perceptual behavior control (PBC) are 9.199, 10.694, 8.629, 4.714, respectively. These numbers are far greater than 1.96, and the p-values are all 0. Therefore, it can be seen that the relevant policies for the National Park System Pilot Area have tremendous potential, which influences community residents' affective commitment and community participation. From large to small, the order of these influences was subjective norms (SN), behavioral attitude (BA), perceptual behavior control (PBC), and behavioral intention (BI). Therefore, the results indicate that the National Park System Pilot Area has a significant direct impact on the behavior and ideology of the community residents and an indirect impact on community residents' behavioral intentions.

The critical value (CR) and the p-value of the behavioral attitude (BA) of the community residents to the behavioral intention (BI) of the community residents are 4.738 and 0,

respectively. Compared with other latent variables, the behavioral attitude has a more significant positive influence on the behavioral intention. Therefore, it can be inferred that the subjective attitude of community residents is the core element that determines the willingness to participate.

The critical value (CR) and the *p*-value of the impact of community residents' subjective norms (SN) on community residents' participation intentions are 3.257 and 0.037, respectively, which correspond with the limit of the correlation coefficient of the positive path influence. The value of the standardized path coefficient is smaller among all of the latent variables, which indicates that the subjective norms of the community residents have limited influence on the participation of community residents. Therefore, it was found that relatives, friends, and community residents' subjective perceptions have no significant influence on an individuals' enthusiasm for participating in constructing the Qilian Mountain National Park.

The critical value (CR) and the *p*-value of the impact of community residents' perceptual behavior control (PBC) on community residents' participation intentions are 2.388 and 0.017, respectively, which supports positive effects. In addition, the standardized path coefficient is 0.89, which is the largest value among all the latent variables. Hence, as the latent variable preset by the model, perceptual behavior control plays a critical role in investigating community residents' participation in constructing national parks. On the other hand, the result reflects the personal will and related abilities of community residents to determine their willingness to participate in the construction of the Qilian Mountain National Park to the largest extent.

## 5. Discussion

The Qilian Mountain National Park System Pilot Area has a significant and positive impact on community residents' participation. It indirectly affects residents' participation intentions through intermediary variables, including behavioral attitude, subjective norms, and perceptual behavior control. The results show that the National Park System Pilot Area has a relatively low impact on the willingness of the community to participate: the related impact system (path coefficient = 0.41) < participation behavior attitude (path coefficient = 0.68) < perceptual behavior control (path coefficient = 0.89). Among the variables, "related impact path–community residents' participation" is a one-way direct effect. The influence effects of "related impact path–participation behavior attitude", "related impact path–subjective norms", and "related impact path–perceptual behavior control" are 0.71, 0.86, and 0.67, respectively.

The results indicate that perceptual behavior control is the dominant factor that affects the participation willingness of community residents in the Qilian Mountain National Park. Specifically, the order of influence is perceptual behavior control (path coefficient = 0.89) > participation behavior attitude (path coefficient = 0.68) > related impact system (path coefficient = 0.41) > subjective norms (path coefficient = 0.38).

The subject norm (path coefficient = 0.38) has the least significant impact on residents' awareness of participation. This indicates that residents in indigenous communities have a constant understanding of the significance of the existence of the Qilian Mountain National Park Pilot Area. Even the relatives and colleagues of these community residents have different perspectives on the issue regarding the community's participation, and most residents will not be affected.

## 6. Conclusions and Implications

Taking the Tianzhu and Sunan counties in the Qilian Mountain National Park area in Gansu province as examples, this research was conducted to investigate the impact of the Qilian Mountain National Park System Pilot Area on community residents' willingness to participate. After a comprehensive analysis, we drew the following conclusions and policy implications.

*6.1. Conclusions*

This section discusses this article's research conclusions and the related research conclusions drawn by previous scholars. This study analyzed the impact of major events related to the National Park System Pilot Area on the psychological perception of community residents and compared the related model to obtain the similarities and differences to related research.

With regard to the impact of significant events on community residents' perceptual intentions, this study's findings are similar to those of Zheng et al. (2014). Specifically, the National Park System Pilot Area serves as a "catalyst," indirectly deepening residents' local emotional identification and promoting individuals' willingness to build a new social network. However, different to previous studies, this research mainly focused on the impact of the three intermediate variables (behavioral attitude, subjective norms, and perceptual behavior control) on the participation intention related to the National Park System Pilot Area. That is, taking the National Park System Pilot Area as the only independent variable, and the participation intention as the only dependent variable, we investigated the community residents' perceptions and willingness to participate in major reforms (e.g., the implementation of the National Park System Pilot Area).

Most scholars have focused on environmental protection and other relevant issues for studies related to the National Park System Pilot Areas and community issues. The focus of this article is the impact on the willingness of community residents to participate. We set the National Park System Pilot Area as the only influencing factor in the preset model. They analyzed the specific relationships among each variable by influencing the intermediate variables. Based on the results, we found that the National Park System Pilot Area has a certain positive impact on community residents' willingness to participate. The findings of this article are consistent with the research conclusions drawn by Zhou et al. (2017) that related to the perceived impact of community participation. In addition, we utilized the theory of planned behavior (TPB) model to subdivide the subjective perceptual factors that affect community participation and clarify the results of different responses of the relevant factors (behavioral attitude, subjective norms, perceptual behavior control) to the dependent variable (the willingness of community residents to participate) when these factors are affected by the independent variable (the National Park System Pilot Area).

*6.2. Implications*

Based on the background that China's national parks are still in an exploratory stage, in order to increase the participation of community residents following the implementation of the Qilian Mountain National Park System Pilot Area, the following implications should be taken into account.

1.   The National Park System Pilot Area positively increased the community residents' willingness to participate. Therefore, regarding future development, it is necessary to strengthen government guidance and popularize scientific research and instructions related to the theme of the Qilian Mountain National Park. In addition, it is critical to ensure that during the exploration period, the pilot policy can reach a critical consensus with the community in the construction of national parks and promote residents' participation intentions in relevant communities through positive relationships.

2.   The National Park System Pilot Area, behavioral attitude, subjective norms, and perceptual behavior control all positively affected community residents' willingness to participate. Therefore, the government may need to consider setting up standardized and professional departments to reward and commend community participants who actively participate in constructing national parks and who demonstrate certain achievements. This method would enhance residents' sense of honor and improve the quality of community life. By "reshaping" the cultural landscape of Qilian Mountain National Park, the community will become the major component of the cultural landscape. The Qilian Mountain National Park cultural value system will be formed based on the principle of attitude priority. In addition, it is necessary to classify the

study areas during the next phase of managing the national park communities and subdivide the relevant suggestions from different communities' feedback regarding the residents' perceptions of participation.

3.  The impact of the construction of the National Park Pilot Area on community residents discussed in this study has a strong correlation with Wallner's three major perceptions of protected areas that affect residents (the economic situation, the history of natural protection, and the power balance between the involved stakeholders) [33]. The authors of this study believe that economic factors and the balance of interests of all parties are essential indicators for coordinating community participation. In addition, we also note that the majority of respondents were not highly educated and had limited abilities to obtain relevant information regarding the development of the national park community. Insufficient publicity of the national park concept and passive acceptance of the policy are the reasons why many respondents have a neutral or negative attitude [34,35]. Therefore, increasing the level of community participation in national parks requires not only active publicity, but also requires relevant organizations to establish effective communication mechanisms and social networks so that residents can actively interact with the National Park Management Committee on issues encountered in the construction of national parks, thereby increasing participation in constructing the national park community.

This study introduces the relevant variables of the Qilian Mountain National Park System Pilot Area into the theory of planned behavior (TPB) model. Based on field surveys and expert interviews, we hypothesized that these variables significantly impact the communities' willingness to participate, and the empirical results supported this hypothesis. At the same time, based on previous research, this study was conducted in economically underdeveloped areas dominated by ethnic minority groups in the national park communities and investigated the impacts of residents' willingness to participate.

**Author Contributions:** Conceptualization, L.J. and J.W.; methodology, J.W.; software, J.W.; validation, L.J., J.W. and Z.W.; formal analysis, L.J.; investigation, J.W.; resources, Z.W.; data curation, J.W.; writing—original draft preparation, J.W.; writing—review and editing, Z.W.; visualization, L.J.; supervision, L.J.; project administration, J.W.; funding acquisition, L.J. All authors have read and agreed to the published version of the manuscript.

**Funding:** This research was supported by the National Natural Science Foundation of China (Grant No. 51968042).

**Institutional Review Board Statement:** The protocol was reviewed and approved by The Lanzhou University of Technology of Science and Technology Board and the Protocol Number is LUT-SAD-51968042.

**Informed Consent Statement:** Not applicable.

**Data Availability Statement:** Some or all data, models, or code that support the findings of this study are available from the corresponding author upon reasonable request.

**Conflicts of Interest:** The authors declare no conflict of interest.

## Notes

1.   The model is extended based on the TPB theoretical model. The extension item is the relevant influence system of the Qilian Mountain National Park Pilot Area. This extension item has a particular impact on cognitive atti-tude and behavioral intention.

2.   The structural equation model is a verification of the model assumptions in Figure 1. The connecting arrows represent the establishment of the influence, and the values under the arrows represent the degree of influence. Therefore, the expected assumptions in Figure 1 are all established.

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
