# Peer review of "The Intention of Community Participation in the Qilian Mountain National Park Policy Pilot"

_land, doi:10.3390/land11020170_

Round 1
Reviewer 1 Report
The manuscript presents very clearly the methodology followed applied to a specific area and with a reduced population sample. The results are presented in a very correct way and could be applied as a methodology for the evaluation and analysis of citizen participation in the face of any territorial action that implies changes in behavior, perception, etc., for that population.
However, the analysis of the participation of a population with special characteristics such as the one that inhabits the territory presented, indigenous, etc ... would the same result be the same if this analysis were applied to populations from sub-Saharan Africa, the United States or Western Europe?
I think that a final reflection should be made in which this situation is valued, depending on the territorial context, the population characteristics (especially in the field of access to secondary and higher education), the ability of the population to access information, the use of social networks or the ability they have to organize in associations that demand rights from the government, among other issues.
-Line 63: Change “In reality” by “Actually” or “In fact”,
-Line 89: Don’t separate “undersand-ing”. It is better: “understan-ding”.
-Line 108: It is better: “resi-dents”.
-Line 117: The figure 1 is “self-elaboration”. I think you should indicate it.
-Line 126: Change “Polit” by “Pilot”.
-Lines 127-138, 147, 155 and 163: You must use the symbols of each paragraph/section as established by the publisher’s rules.
-Line 142: Change by Re-cognaize.
-Section 3.1. I think that a map of the location of these areas must be included, especially for those lectors that do not know the region/country of China. It will improve the international relevance of the manuscript.
-Line 186: You indicate “see table 2”, but there is no table 2 in the manuscript. There are table 2-1, 2-2, 2-3, but I cannot find table 2 where the likert 5 scale is indicated. Perhaps the problem is that instead of presenting a table 2, you present a table 2-1 and 2-2, when they really refer to table 2 as a whole. I think that to solve the problem it is better to number both tables as Table 2, and if part of it has to appear on a different page, name it with the same title: "Table 2 (continued) research aspects and measurement items". Always complying with the rules established by the publisher. The same suggestion for Tables 3-1 and 3-2.
-Line 234: be careful in the final version: Don’t separate the title from the table (in different pages, I mean).
-Table 1: Percentages of male and female are wrong. You must correct female to 47,2%.
-Table 2-1: You should also indicate in the 4rd column the number of the references, that corresponds with the section “References” at the end of the manuscript. Example: Zhang et al. (2017) [20].
You should also follow these indications for other references within the text (it has been indicated with comments in the attached file).
The articles of the references 18 and 25, must follow the rules of the publisher. They must not be in capitals but in lowercase.

Author Response
Response to Reviewer 1 Comments
Point 1: the analysis of the participation of a population with special characteristics such as the one that inhabits the territory presented, indigenous, etc ... would the same result be the same if this analysis were applied to populations from sub-Saharan Africa, the United States or Western Europe?
Response 1: The research method can be applied to Africa, Western Europe and other regions, but it needs to know the local policy and culture, especially for the design of questionnaire items, it needs to analyze the prominent problems in the development of local national parks and cultural background, because the definition of community participation may be different in various countries, which will affect the existing research conclusions in some ways. Therefore, the wide applicability of the research will be explored in the future.
Point 2: I think that a final reflection should be made in which this situation is valued, depending on the territorial context, the population characteristics (especially in the field of access to secondary and higher education), the ability of the population to access information, the use of social networks or the ability they have to organize in associations that demand rights from the government, among other issues.
Response 2: The suggestion had been supplemented in the discussion section of this paper. Accroding to line XXX to XXX, The impact of the construction of the national park pilot area on community residents discussed in this study has a strong correlation with Wallner’s three major perceptions of protected areas that affect residents (the economic situation, the history of nature protection, and the power balance between the involved stakeholders). Researchers of this study believe that economic factors and the balance of interests of all parties are essential indicators in coordinating community participation. In addition, the researchers also noted that the majority of the respondents were not highly educated and had limited abilities to obtain relevant information regarding the development of the national park community. Insufficient publicity of the national park concept and passive acceptance of the policy are the reasons why many respondents have a neutral or negative attitude. Therefore, increasing the level of community participation in national parks requires not only active publicity but also requires relevant organizations to establish effective communication mechanisms and social networks so that residents can actively interact with the National Park Management Committee on issues encountered in the construction of national parks, thereby increasing the participation in the constructing the national park community.
Point 3:
-Line 63: Change “In reality” by “Actually” or “In fact”,
-Line 89: Don’t separate “undersand-ing”. It is better: “understan-ding”.
-Line 108: It is better: “resi-dents”.
-Line 117: The figure 1 is “self-elaboration”. I think you should indicate it.
-Line 126: Change “Polit” by “Pilot”.
-Lines 127-138, 147, 155 and 163: You must use the symbols of each paragraph/section as established by the publisher’s rules.
-Line 142: Change by Re-cognaize.
-Line 186: You indicate “see table 2”, but there is no table 2 in the manuscript. There are table 2-1, 2-2, 2-3, but I cannot find table 2 where the likert 5 scale is indicated. Perhaps the problem is that instead of presenting a table 2, you present a table 2-1 and 2-2, when they really refer to table 2 as a whole. I think that to solve the problem it is better to number both tables as Table 2, and if part of it has to appear on a different page, name it with the same title: "Table 2 (continued) research aspects and measurement items". Always complying with the rules established by the publisher. The same suggestion for Tables 3-1 and 3-2.
-Line 234: be careful in the final version: Don’t separate the title from the table (in different pages, I mean).
-Table 1: Percentages of male and female are wrong. You must correct female to 47,2%.
-Table 2-1: You should also indicate in the 4rd column the number of the references, that corresponds with the section “References” at the end of the manuscript. Example: Zhang et al. (2017) [20].
You should also follow these indications for other references within the text (it has been indicated with comments in the attached file).
The articles of the references 18 and 25, must follow the rules of the publisher. They must not be in capitals but in lowercase.
Response 3: The above questions had been revised in the latest article
Point 4: -Section 3.1. I think that a map of the location of these areas must be included, especially for those lectors that do not know the region/country of China. It will improve the international relevance of the manuscript.
Response 4: Qilian Mountain National Park, Sunan County and Tianzhu County had been marked on the map through GIS software. (See Figure 2 for details in this paper)

Reviewer 2 Report
Review of land-1536729
Line 55: ‘in conclusion’ is a wrong choice of word here
Methodology: The sampling frame, survey administration and response rate are inadequately described.
Table 1 mistake in the percentages for gender, attribute of residents
Line 240-241 “The statistical results show that the proportion of males and females in the survey subjects is the same” This statement is incorrect. 120 males vs 110 females.
Given the lack of clarity related to sampling procedures it is futile to assess the rest of the paper.
Also, I am VERY concerned that there is no indication that the authors obtained approval from either of the academic institutions they are affiliated with. In this day and age, ethics approval must be obtained BEFORE the study starts. And the approval, incl. the protocol number must be stated in the article. As long as this is not provided, the paper must be rejected.
On a formal level, the paper needs an edit by a NATIVE English speaking professional editor. It is full of infelicities of expression and grammar.
Author Response
Response to Reviewer 2 Comments
Dear editor,
We appreciate your feedback. Please see the attachment, which is the modified version of our article, and the answers below:
Point1: Line 55: ‘in conclusion’ is a wrong choice of word here?
Response 1: we deleted the phrase “in conclusion” in the line 55.
Point2: Methodology: The sampling frame, survey administration and response rate are inadequately described.
Response 2: Based your recommendation, we supplemented information regarding the detailed sampling procedures, survey administration and response rate. Please see line 257-283 for details.
Point3: Table 1— mistake in the percentages for gender, attribute of residents
Response 3: The calculated value has been modified to 47.8%
Point4: Line 240-241 “The statistical results show that the proportion of males and females in the survey subjects is the same” This statement is incorrect. 120 males vs 110 females.
Response 4: The corresponding content has been modified.
Point5: I am VERY concerned that there is no indication that the authors obtained approval from either of the academic institutions they are affiliated with. In this day and age, ethics approval must be obtained BEFORE the study starts. And the approval, incl. the protocol number must be stated in the article. As long as this is not provided, the paper must be rejected.
Response 5: In the early stage of the research, we had obtained the ethical recognition of the science and Technology Department of LUT. Please see attachment 1 for the approved letter in the end of paper. It is an official letter provided by “Lanzhou University of Technology”.
Point6: On a formal level, the paper needs an edit by a NATIVE English speaking professional editor. It is full of infelicities of expression and grammar.
Response 6: We submitted the manuscript to the MDPI. They helped us correct the grammar issues. Please see attachment 2 for the English-Editing Certification in the end of paper.

Reviewer 3 Report
The article is well structured and fundamented. It introduces some innovation regarding the case studies and hyphotesis tested. Results and analyisis demonstrate relevant results for future research on community participation..
Author Response
Response to Reviewer 3 Comments
Dear editor,
We appreciate your feedback. We submitted the manuscript to the MDPI. They helped us correct the grammar issues. Please see attachment for the English-Editing Certification in the end of paper.

Round 2
Reviewer 2 Report
The authors have addressed my of my concerns
Author Response
Dear editor,
We appreciate your feedback.And we have revised the relevant questions you raised, especially to correcting the grammar issues.
